# Systematic profiling of conditional degron tag technologies for target validation studies

Daniel P. Bondeson[1], Zachary Mullin-Bernstein[1], Sydney Oliver[1], Thomas A. Skipper[1], Thomas C. Atack [1], Nolan Bick[1], Meilani Ching[1], Andrew A. Guirguis [1,2,3,4], Jason Kwon[1,2], Carly Langan[1], Dylan Millson[1,2], Brenton R. Paolella [1], Kevin Tran[3,4], Sarah J. Wie[1], Francisca Vazquez[1], Zuzana Tothova[1,2], Todd R. Golub [1,2], William R. Sellers [1,2,5] & Alessandra Ianari[1,5] ✉

Conditional degron tags (CDTs) are a powerful tool for target validation that combines the kinetics and reversible action of pharmacological agents with the generalizability of genetic manipulation. However, successful design of a CDT fusion protein often requires a prolonged, ad hoc cycle of construct design, failure, and re-design. To address this limitation, we report here a system to rapidly compare the activity of five unique CDTs: AID/AID2, IKZF3d, dTAG, HaloTag, and SMASh. We demonstrate the utility of this system against 16 unique protein targets. We find that expression and degradation are highly dependent on the specific CDT, the construct design, and the target. None of the CDTs leads to efficient expression and/or degradation across all targets; however, our systematic approach enables the identification of at least one optimal CDT fusion for each target. To enable the adoption of CDT strategies more broadly, we have made these reagents, and a detailed protocol, available as a community resource.

A major focus in biomedical research is the discovery of novel therapeutic targets. Many targets arise from genome-wide association studies or large-scale functional experiments[1,2] and are often poorly characterized, bringing exciting opportunities and concomitant challenges. When the underlying biological functions of a novel protein of interest (POI) are unknown, the consequences of its perturbation are also unpredictable. These unknowns are relevant for estimating both therapeutic efficacy and potential normal tissue toxicity. Unfortunately, the generation of tool compounds to enable proof-of-concept efficacy and safety studies requires significant time and resources and such compounds are not available early in target validation studies, precisely when the need to address these questions is most important[3,4].

Genetic suppression strategies, including CRISPR/Cas9 and shRNA are valuable and generalizable tools for target validation;

however, these modalities suffer from off-target effects (especially for shRNA[5,6]), slow mechanisms of action, irreversible perturbation of the POI (CRISPR), and an inability to titrate the extent of POI inhibition. As such, these technologies often do not recapitulate the phenotypic consequences seen with a small molecule or biologic inhibitor. To address these concerns, chemical genetic systems for post-translational protein control have been developed, which we refer to as conditional degron tags (CDTs). These systems employ protein tags rendering a fused-protein partner sensitive to tunable protein degradation upon treatment with a "degrader drug"[7–16]. Importantly, CDTs can impart rapid degradation kinetics, reversible recovery of protein levels after drug removal, and orthogonal and/or known off-target effects of the degrader drug.

Although CDTs show promise as powerful tools for early target validation, their generalizability across many POIs has not been

[1]The Broad Institute of MIT and Harvard, Cambridge, MA, USA. [2]Dana-Farber Cancer Institute, Harvard Medical School, Boston, MA, USA. [3]Peter MacCallum Cancer Centre, Melbourne, VIC, Australia. [4]Sir Peter MacCallum Department of Oncology, The University of Melbourne, Melbourne, VIC, Australia. [5]These authors contributed equally: William R. Sellers, Alessandra Ianari. ✉e-mail: aleianari12@gmail.com

evaluated systematically. Anecdotally, we have found highly variable results when applying these different CDTs to cancer-relevant targets, which significantly impacted our ability to use CDTs for our studies. Two possibilities are presented: either certain CDTs can be broadly used against various POI with robust activity and should be prioritized, or each POI requires a unique strategy, thus requiring a systematic assessment of multiple CDT strategies.

Here, we evaluate the robustness and generalizability of five different CDTs fused to either the N- or C-terminus of 16 different POIs and expressed as V5 fusions under a strong (SFFV) or a weak (PGK) promoter. Importantly, we find that none of the CDTs analyzed performed optimally across all targets. Instead, we find that expression levels and drug-induced degradation varied widely and in an unpredictable manner. Still, our systematic analysis of different CDT fusions allows the identification of potently degraded CDTs for each POI. Finally, we compare multiple CDTs in functional assays for two POIs, and in each case, we identify at least one CDT fusion protein that

phenocopied both expression of the wild-type protein and its genetic inactivation. Together, these results indicate that parallel testing of multiple CDTs enables the rapid and successful development of POI-CDT fusions for target validation studies.

## Results

### Generation of a systematic CDT panel of lentiviral vectors

We focused on five CDTs (Fig. 1a): auxin-inducible degron (AID)[8,9], dTAG[10,11], IKZF3 degron (aa130–189, IKZF3d)[12,13], HaloTag[14,15], and small molecule-assisted shut-off (SMASh)[16]. Except for the SMASh tag, each CDT achieves degradation by "reprogramming" an E3 ubiquitin ligase with a small molecule to recognize the CDT-tagged protein. In contrast, SMASh-tagged fusion proteins are processed by the proteolytic self-cleavage of a degron to yield the untagged, wild-type POI; here, treatment with protease inhibitors allows the expression of the degron-fused POI—which is rapidly degraded—and the gradual disappearance of the untagged POI. We generated a panel of 20 lentiviral

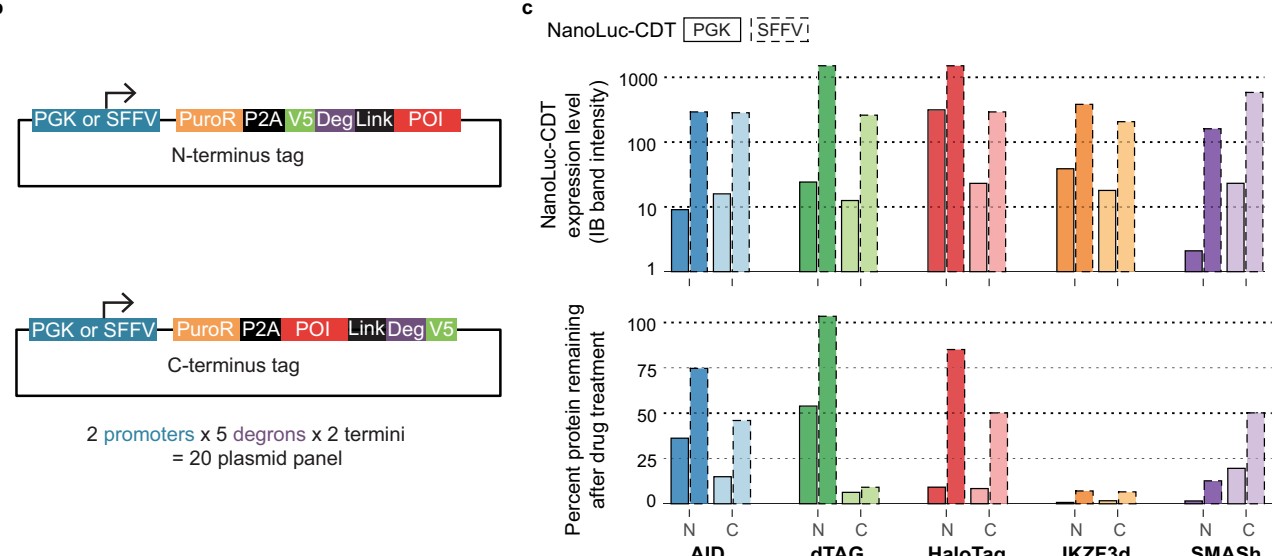

**a**

| Conditional Degron Tag (CDT) | Size | Mechanism | Drug and top dose tested | Ref. |
|---|---|---|---|---|
| Auxin Inducible Degron (AID) | 7 kDa | Molecular glue for the Tir1 E3 ligase | Indole-3-acetic acid (IAA, 500 µM) | 8, 9 |
| dTAG | 12 kDa | PROTAC for the CRBN or VHL E3 ligases | dTAG13 or dTAG V1 (1 µM) | 10, 11 |
| IKZF3d | 7 kDa | Molecular glue for the CRBN E3 ligase | Pomalidomide (Pom, 1 µM) | 12, 13 |
| HaloTag | 34 kDa | PROTAC for the VHL E3 ligase | HaloPROTAC3 (1 µM) | 14, 15 |
| SMASh | 7* kDa | Prevents self-cleavage of unstable degron | Asunaprevir (1 µM) | 16 |

**b** Plasmid design: N-terminus tag and C-terminus tag.
2 promoters x 5 degrons x 2 termini = 20 plasmid panel

**c** NanoLuc-CDT (PGK / SFFV)

**Fig. 1 | A vector panel to systematically evaluate conditional degron tag (CDT) fusion protein expression, degradation, and function. a** Summary table of the five CDTs evaluated in this work. Each CDT, except for small-molecule-assisted shut-off (SMASh), functions through reprogramming an E3 ubiquitin ligase with a small molecule to recognize the degron fusion protein. SMASh fusion proteins self-cleave a degron tag; the addition of the small molecule blocks this self-cleavage. *The SMASh tag itself is 34 kDa, but after self-cleavage, the stably-expressed (V5) tag is only 7 kDa. **b** Plasmid design for systematically evaluating CDT fusions to a protein of interest (POI). We employed a P2A cleavage site to separate an antibiotic-resistance gene (PuroR) and the fusion protein. A V5 epitope tag enables detection and quantitation of the fusion protein by western blot. A rigid linker (EAAK₃, Link)

was also incorporated between the degron tag and the cloning site in all constructs except AID and SMASh, where the CDT is separated from the POI by the V5 tag. These expression cassettes were cloned into a lentiviral vector for stable integration into the genome. **c** Validation of the CDT panel using NanoLuciferase. Nano-Luciferase fused to the indicated CDT was expressed using the PGK or SFFV plasmids. Here, expression levels are reported as the anti-V5 IB band intensity normalized to the anti-Vinculin band intensity and then scaled to the total expression of all ten CDTs evaluated to highlight the relative expression differences between the different CDTs. Degradation is reported as the percent reduction in the anti-V5 IB band intensity after drug treatment. See Supplementary Fig. 2a for the full IB for both PGK and SFFV constructs.

vectors to enable testing any POI with these different CDTs fused at either terminus, expressed under the control of a weak (PGK) or strong (SFFV) promoter (Fig. 1b). We hypothesized that the resulting panel of expression vectors would enable the more rapid discovery of at least one degradable construct for any given POI.

To validate the vector design, we cloned NanoLuciferase (NanoLuc[17]) into each vector in-frame with the CDT. Polyclonal HEK-293T cells stably expressing each NanoLuc-CDT construct were generated via lentiviral transduction under conditions favoring single integration to allow for the comparison of expression levels. To assess AID tags, we co-expressed the *Oryza sativa* E3 ubiquitin ligase Tir1, as reported previously[9]. To efficiently assess degradation across different doses (as previously reported[9,10,12,14,16], 10 nM to 1 μM for all drugs except indole-3-acetic acid (IAA), which was tested at 5 to 500 μM) and time points (6–72 h), we developed an anti-V5 in-cell western (ICW, Supplementary Fig. 1a–d). Except for minor viability defects observed for HaloPROTAC3 at 1 μM, these doses were not toxic to HEK-293T cells (Supplementary Fig. 1a). We evaluated the baseline expression of each NanoLuc-CDT fusion via anti-V5 immunoblot (IB, Fig. 1c and Supplementary Fig. 2a). We noted different expression levels for each construct and generally higher expression from the SFFV promoter compared to the PGK promoter. We further confirmed the degradation of the NanoLuc-CDT fusions by immunoblotting (Supplementary Fig. 2a) and found that these constructs were degraded with various efficiencies, ranging from <50% (dTAG-N) to near complete degradation (e.g., IKZF3d and dTAG-C, Fig. 1c, bottom and Supplementary Figs. 1c, 2a).

## CDT Performance across 16 unique POIs

We next performed similar experiments testing a total of 16 unique POIs, spanning different protein classes including ten cytoplasmic proteins (Green Fluorescent Protein (GFP), NanoLuciferase (NanoLuc), Red-shifted Firefly Luciferase (RFluc[18]), Protein arginine methyltransferase 2 (PRMT5[19]), WD Repeat and SOCS Box-Containing 2 (WSB2), induced myeloid leukemia cell differentiation protein (MCL1[20]), Soc-2 Suppressor Of Clear Homolog (SHOC2[21,22]), Methionine Adenosyltransferase 2 A (MAT2A[23]), Vacuolar protein sorting-associated protein 4 (VPS4A[24]), and Protein Activator of interferon-induced Protein Kinase EIF2AK2 (PRKRA[25]); three nuclear proteins (Stromal Antigen 1 and 2 (STAG1 and STAG2[26]), DNA (cytosine-5)-methyltransferase 1 (DNMT3A[27]); two multi-pass plasma membrane proteins (Xenotropic and Polytropic Receptor 1 (XPR1[28,29]), and Kinase D interacting protein of 220 kDa (KIDINS220[30,31]); and one single-pass transmembrane receptor tyrosine kinase (Fms-Related receptor tyrosine kinase 3 with the internal tandem duplication, FLT3-ITD[27]).

We generated polyclonal HEK-293T cell populations stably expressing each POI-CDT fusion from either the PGK or SFFV promoter (as noted) and assessed protein expression and degradation by IB and ICW (Fig. 2a and Supplementary Figs. 2–6). We found that expression levels were highly variable across different constructs and on a target-by-target basis. Some CDT fusions were poorly expressed or undetectable by IB (e.g., see arrows in Supplementary Figs. 2–6). In one case (WSB2 expressed from the SFFV promoter, Supplementary Fig. 6b), CDT fusions were detectable only when cloned at the C-terminus. In general, we noted that N-terminal dTAG and HaloTag fusions produced the highest levels of the full-length protein (Fig. 2a and Supplementary Fig. 7a).

We next evaluated the extent of degradation of each CDT fusion protein (Fig. 2a and Supplementary Figs. 2–6). Here, the maximal degradation observed varied greatly based on both the target and CDT fusion strategy (i.e., N- vs C-terminus). Some targets were highly amenable to degradation with almost every CDT (e.g., VPS4A, PRKRA, and PRMT5), while other targets were resistant to degradation with most constructs (e.g., only N-terminal dTAG, IKZF3d, and SMASh fusions were degraded for KIDINS220, Supplementary Fig. 5c). We,

therefore, asked whether there were generalizable trends that might favor choosing one CDT strategy over others.

In some cases, we predict that high levels of expression prohibit efficient degradation. This is most relevant for the N-terminal dTAG and HaloTag fusions which were expressed at the highest level across targets (Fig. 2a and Supplementary Fig. 7a). Indeed, NanoLuc-HaloTag-N was degraded by 91% when expressed from the PGK promoter, but only by 15% when expressed from the SFFV promoter (Fig. 1c). We observed the same phenomena with forced overexpression of the same construct using transient transfection (e.g., PRMT5-dTAG-N, Supplementary Fig. 7b, c) or stable expression using different viral titers (XPR1-HaloTag-N, Supplementary Fig. 7d). These data suggest that high levels of overexpression may overwhelm protein homeostatic mechanisms leading to, for example, a saturation of the degradation machinery or protein misfolding, yielding a non-ligandable target. Nevertheless, titrating expression using a weaker promoter may overcome this bottleneck.

In general, the AID system was expressed poorly and only minimally degraded; the dynamic range of protein levels was the smallest of all five CDTs. This has been documented previously[8,32,33], and an AID2 system has been developed to address these limitations[8]. Therefore, we compared AID and AID2 performance across nine different targets (NanoLuc, RFluc, GFP, MAT2A, MCl1, WSB2, PRMT5, XPR1, and KIDINS220, Fig. 3b and Supplementary Fig. 8) and confirmed that the AID2 system indeed led to a small but consistent increase in basal protein levels, up to a threefold increase for KIDINS220 (Supplementary Fig. 8i, j). In addition, degradation potency for moderately degraded targets with AID was improved in the AID2 system (Supplementary Fig. 8k). For PRMT5, AID-N showed 85% degradation with AID2, while the original AID system only achieved a ~5% reduction in protein levels (Supplementary Fig. 8f). Thus, for some targets, the AID2 system would be a valuable CDT.

With these considerations in mind, we found that dTAG and SMASh generally provide the best dynamic range between baseline expression levels and efficiency of degradation across 12 (75%) of the targets analyzed (Fig. 2c, d). Although IKZF3d constructs were degraded robustly across the same number of targets, the poor expression of IKZF3d fusion proteins decreases the overall dynamic range. The HaloTag system enabled degradation for a smaller number of targets, and, as noted above, only a few targets were degraded efficiently by the AID/AID2 systems. Nevertheless, several POI were only efficiently degraded by these less robust technologies (e.g., XPR1 with AID2-C). Notably, while none of the CDTs tested worked across all targets, testing multiple CDTs and fusion strategies in parallel allowed us to consistently identify at least one CDT fusion construct for each POI that was efficiently expressed and degraded (>95%) in the presence of the relative degrader drug (Fig. 2c, d).

## Kinetics of degradation and recovery across CDT fusion proteins

A key feature of CDTs is the ability to modulate the expression of a given target with fast and reversible kinetics. To determine how different CDTs performed in this regard, we compared the kinetics of degradation for each CDT across multiple targets. While each CDT has an exemplary POI with complete degradation after 24 h of drug exposure, the specific POI had a large effect on the kinetics of degradation (Fig. 3 and Supplementary Fig. 9, 10). For example, GFP and RFLuc fusion proteins were degraded rapidly by most CDTs within 6 h of treatment, while maximal degradation of the transmembrane protein XPR1 was only observed after 48 h (e.g., XPR1-dTAG, Fig. 3b). These findings suggest that certain substrates might have fast re-synthesis rates that must be overcome by the optimal CDT, engage more slowly with the degradation machinery, and/or require additional processing before protein levels decrease (e.g., extraction from the plasma membrane). In general, we observed slower kinetics of degradation for the SMASh tag fusion proteins, consistent with its

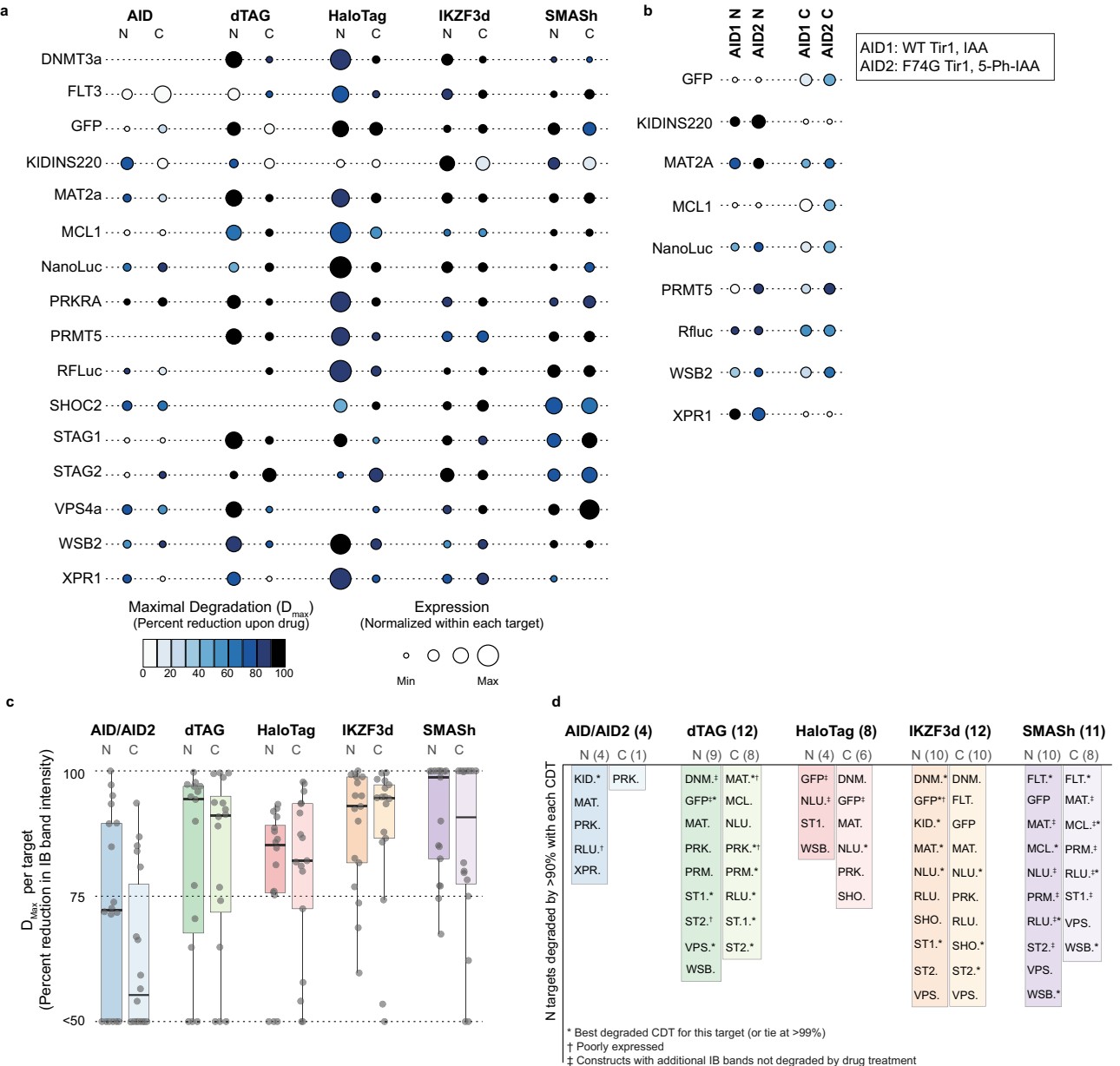

**Fig. 2 | Comparison of degradation efficiency for five CDTs across 16 unique proteins of interest. a** Expression and degradation across CDT fusions to 16 unique proteins of interest (POI). The indicated POI-CDT fusions were expressed in HEK-293T cells treated with DMSO or the respective degrader drug and analyzed by immunoblot (IB, see Supplementary Figs. 2–6). The size of each dot corresponds to the baseline expression level of that fusion protein: the anti-V5 IB band intensity normalized to the anti-Vinculin band intensity and then scaled to the total expression of all 10 CDTs evaluated for a given target to highlight the relative expression differences between the different CDTs. The color of each point indicates the maximal degradation observed ($D_{max}$) for that protein after degrader drug treatment. Note that missing data points indicate that the fusion protein was not evaluated and that some of the SMASh-tag constructs were efficiently degraded at later timepoints, as

shown in Supplementary Figs. 9, 10. **b** Comparison of AID and AID2 systems for nine POI. The indicated AID fusions were expressed in HEK-293T cells co-expressing Tir1 (either wildtype for AID and or F74G for AID2). Cells were treated with drug (Indole-3-acetic acid for AID or 5-phenyl-indole-3-acetic acid for AID2), and protein levels were analyzed by immunoblot (see Supplementary Fig. 8). Protein expression is normalized as in **a**. **c** The $D_{max}$ observed for each fusion protein categorized by degron technology and fusion terminus. The box and whisker plot indicates the median, the first and third quartiles, and 1.5x the interquartile range. $N = 16$ targets analyzed for each construct except dTAG-N, where $N = 15$. **d** POI-CDT fusions that were degraded by >95%. NLU. NanoLuc, RLU. RFluc, PRM. PRMT5, WSB. WSB2, MCL. MCL1, SHO. SHOC2, MAT. MAT2A, VPS. VPS4A, PRK. PRKRA, ST1. STAG1, ST2. STAG2, DNM DNMT3A, XPR. XPR1, KID. KIDINS220, FLT. FLT3-ITD.

unique mechanism of action (Supplementary Figs 9, 10). Between different targets, the kinetics ranged from clearance of both the untagged POI and the degron-tagged POI within 24 h (VPS4A, Supplementary Fig. 10c) to 5 days (PRMT5 and WSB2, Supplementary Fig. 10b, d) to retention of the degron-tagged POI for up to 10 days (MCL1-SMASh-C, Supplementary Fig. 9d).

We next evaluated the reversibility of each technology by performing drug washout experiments for the NanoLuc-, RFLuc-, and

PRMT5-CDTs fusions (Fig. 3c and Supplementary Fig. 10e, f). Twenty-four hours after drug treatment, media were replaced with drug-free media and protein levels were monitored for up to 96 h. Except for NanoLuc-dTAG-C, the levels of most CDT fusion proteins recovered within 24 h. Altogether, these results indicate that the kinetics of degradation and recovery are highly variable, unpredictable, and dependent both on the CDT and the POI analyzed and thus, should be determined empirically for each POI fusion.

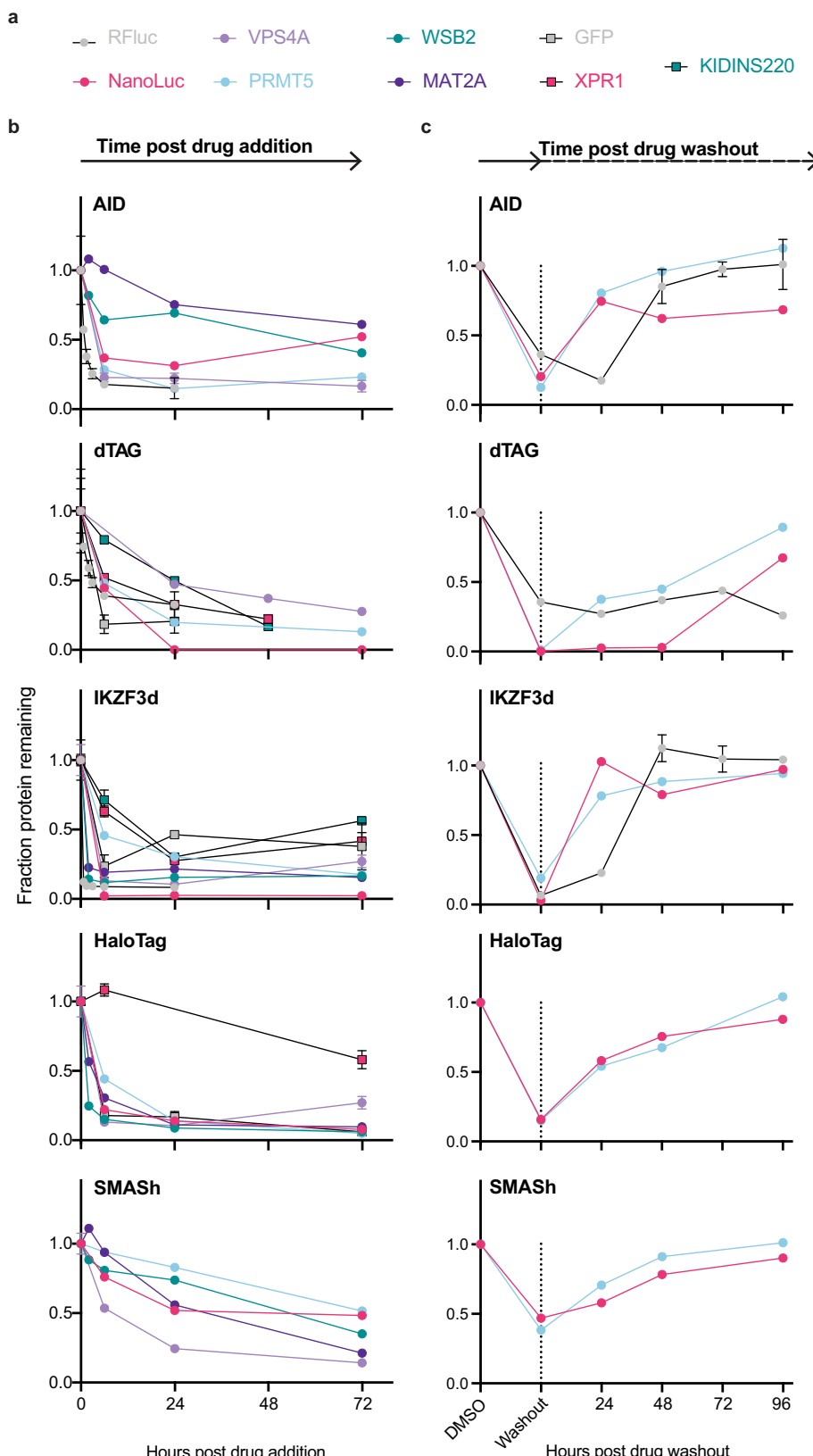

## Functional assessment of XPR1- and MCL1-CDTs

Finally, we assessed the expression and induced degradation in multiple cellular contexts and determined the degree to which a CDT fusion retains the physiological activity of the POI. We observed that for PRMT5-CDT fusion proteins, most cell lines showed similar patterns of expression and degradation (Supplementary Fig. 11a).

However, XPR1-CDT fusion proteins were more variable across different cell lines (Fig. 4a and Supplementary Fig. 11b, c). For example, XPR1-dTAG-N fusion proteins were expressed and degraded in HEK-293T and SNGM cell lines, but barely detectable and not degraded in IGROV1 (Fig. 4a and Supplementary Fig. 11b), possibly due to the important role of XPR1 in maintaining the viability of this cell line[30], or

**Fig. 3 | The kinetics of degradation and fusion protein recovery are highly heterogeneous across different technologies. a** Color legend for the genetic targets profiled in this figure. See Supplementary Fig. 9a for more details on the exact constructs profiled in Fig. 3. **b** Kinetics of degradation after treatment with degrader drugs. After the indicated times, RFluc protein levels were measured via in-cell luciferase signal, while all other protein levels were measured via ICW. Relative protein levels are presented as a fraction of the untreated sample. Errors bars represent the mean and standard error of $N = 2$ separate wells, representative of $N = 2$ independent experiments. **c** Kinetics of recovery after degrader drug washout. After drug treatment (all constructs except SMASh, 24 h; SMASh, 5 days), cells were washed and incubated in a cell culture medium without the drug for the indicated times. RFluc protein levels were measured via in-cell luciferase signal, and error bars represent the standard error of the mean. NanoLuc and PRMT5-CDT protein levels were measured by IB, and data represent the V5 signal normalized to the Vinculin loading control. Errors bars represent the mean of $N = 2$ technical replicates, representative of $N = 2$ independent experiments.

to differences in protein homeostasis mechanisms across cellular contexts[34]. We also assessed the performance of each CDT in murine contexts using the NIH-3T3 cell line. As previously reported, the IKZF3d system was inefficiently degraded in this context while each other CDT maintained the performance observed in HEK-293T (Supplementary Fig. 11d).

Consistent with prior studies, we found that each CDT technology did not impair the enzymatic activity of RFluc (Supplementary Fig. 12a and refs. [9], [10], [12], [15], [16], [35–38]). Thus, we evaluated the ability of two CDT fusion proteins—XPR1 and MCL1—to phenocopy both full activity at baseline and its loss of function upon drug treatment.

The activity of XPR1, the only annotated cellular phosphate exporter in the human genome[29,30], can be assessed using $^{32}$P-labeled phosphate pulse-chase experiments to quantify cellular phosphate export. *XPR1* inactivation (XPR1-KO) significantly decreases cellular phosphate export (Fig. 4b and Supplementary Fig. 12b–d). This phenotype can be rescued by re-expression of wild-type *XPR1* (XPR1-WT) but not by hypomorphic alleles (e.g., XPR1 with the L218S mutation)[39]. With this robust model in hand, we tested the re-expression of five CDT constructs in an XPR1-KO background. Here, we found that all CDT constructs fully restored phosphate efflux activity, often at higher levels relative to XPR1-WT, likely due to high expression (Supplementary Figs. 5b, 12c). Importantly, phosphate efflux activity was significantly decreased in each CDT fusion after treatment with the drug, with HaloTag-N displaying the largest dynamic range and fully recapitulating the XPR1-KO phenotype (Fig. 4b).

As a second exemplary case, we tested the ability of CDT fusion proteins to phenocopy the activity of the antiapoptotic protein MCL1. MCL1 plays a key role in suppressing the activation of the intrinsic apoptotic cascade through its interactions with BH3 proapoptotic proteins (e.g., BIM, PUMA, or NOXA). Genetic inactivation of MCL1 (MCL1 KO) sensitizes cells to the BCL-2, BCL-xL, and BCL-W inhibitor Navitoclax[40]. We tested two MCL1-CDT fusions for their ability to protect cells against Navitoclax treatment—phenocopying endogenous MCL1—and to sensitize cells to Navitoclax upon degradation—phenocopying MCL1 inactivation. Expression of MCL1-SMASh-N failed to rescue the MCL1 KO phenotype when expressed from a weak (PGK) promoter (Fig. 4c, d) but provided a large dynamic range of Navitoclax sensitivity between DMSO and Asunaprevir treatment when expressed from a strong (SFFV) promoter. In contrast, the expression of MCL1-dTAG-C from the stronger promoter impeded complete degradation and cells retained resistance to Navitoclax (i.e. MCL1 activity) even upon s treatment (Fig. 4c, d). However, when expressed at lower levels by the weaker promoter, there was a profound difference in Navitoclax sensitivity between DMSO and dTAG$^V$−1 treatment. These results highlight the importance of carefully evaluating expression and functionality levels for each CDT fusion to yield a useful tool for further target validation experiments.

## Discussion

There is substantial interest in the scientific community to leverage CDTs for target validation studies, and multiple potent CDTs have been developed over the past few years[7]. Their applicability, however, remains complicated by the fact that CDTs do not work consistently across protein targets. In our experience, each target requires iterative and lengthy cycles of cloning and testing to identify an ideal CDT

strategy that leads to sufficient activity and degradation of the CDT-POI fusion to phenocopy the action of a drug.

Here, we asked whether one CDT platform typically outperforms the others, or if testing multiple strategies is necessary to develop a functional CDT. To address this question, we generated a panel of lentiviral vectors to systematically compare the efficiency of five unique CDTs fused to either the N- or C-terminus of 16 unique POIs representing different protein classes. We found that total protein expression and drug-induced degradation was highly dependent on both the technology used and the specific construct design on a target-by-target basis. In comparing efficiencies across CDTs (Fig. 2b), we noticed that the most consistently degraded CDTs were the dTAG, IKZF3d, and SMASh systems. However, it should be noted that dTAG fusions were often expressed at higher levels than SMASh and IKZF3d fusions (Supplementary Fig. 7a). This finding is important as the levels of the CDT fusion can matter both in terms of its degradation potency and in terms of its activity - that is, the ability of the CDT fusion protein to phenocopy the function of the endogenous POI. In addition, having good baseline expression with >95% degradation upon treatment provides a larger dynamic range to study POI activity, which makes for a more robust tool.

Notably, although dTAG and IKZF3d systems generally performed best, certain targets could only be degraded efficiently by other CDTs. For example WSB2 and MCL1 were degraded efficiently (>95%) only by the SMASh-N CDT, whereas the phosphate exporter XPR1, XPR1-HaloTag-N was the only CDT fusion that could be sufficiently degraded to phenocopy *XPR1* inactivation. Interestingly, AID fusions were expressed at lower levels and degraded less efficiently than other CDTs across almost all POIs, with marginal improvements using the AID2 system[8].

We extended our studies to the kinetics of degradation of the various CDTs. Apart from the SMASh fusions, which appeared to have slower kinetics across targets, other CDT fusions showed great variability on a target-dependent basis. It is important to note that in the SMASh system, the removal of the POI is dependent on two variables (1) the half-life of the POI (which will determine how quickly the untagged, cleaved form will disappear) and (2) the kinetics of degradation of the degron-tagged POI, a much higher molecular weight protein with unknown functional properties, which we found to often have very long and/or incomplete kinetics of degradation (Supplementary Figs. 9, 10). Overall, we observed fast kinetics of recovery across CDT fusions, further supporting the notion that CDTs are powerful systems for cell-based assessment of conditional and reversible removal of any given protein.

Finally, we illustrate the importance of assessing the functional activity of POI-CDT fusions when employing a CDT strategy. Certain tags may not be amenable to particular cellular contexts: for example, IKZF3d tags are not degraded efficiently in murine models (Supplementary Fig. 11d) and the degrader drug—pomalidomide—degrades additional targets beyond just the tagged-POI[13]. In addition, the AID tag additionally requires co-expression of the TIR1 E3 ligase. These considerations can be predicted a priori, but the optimal CDT for ensuring a large dynamic range of POI activity is more difficult to predict. We present two case studies—XPR1 and MCL1—to show how the CDT fusion strategy could be hampered if the activity or expression levels are insufficient to phenocopy those of the endogenous protein, or if

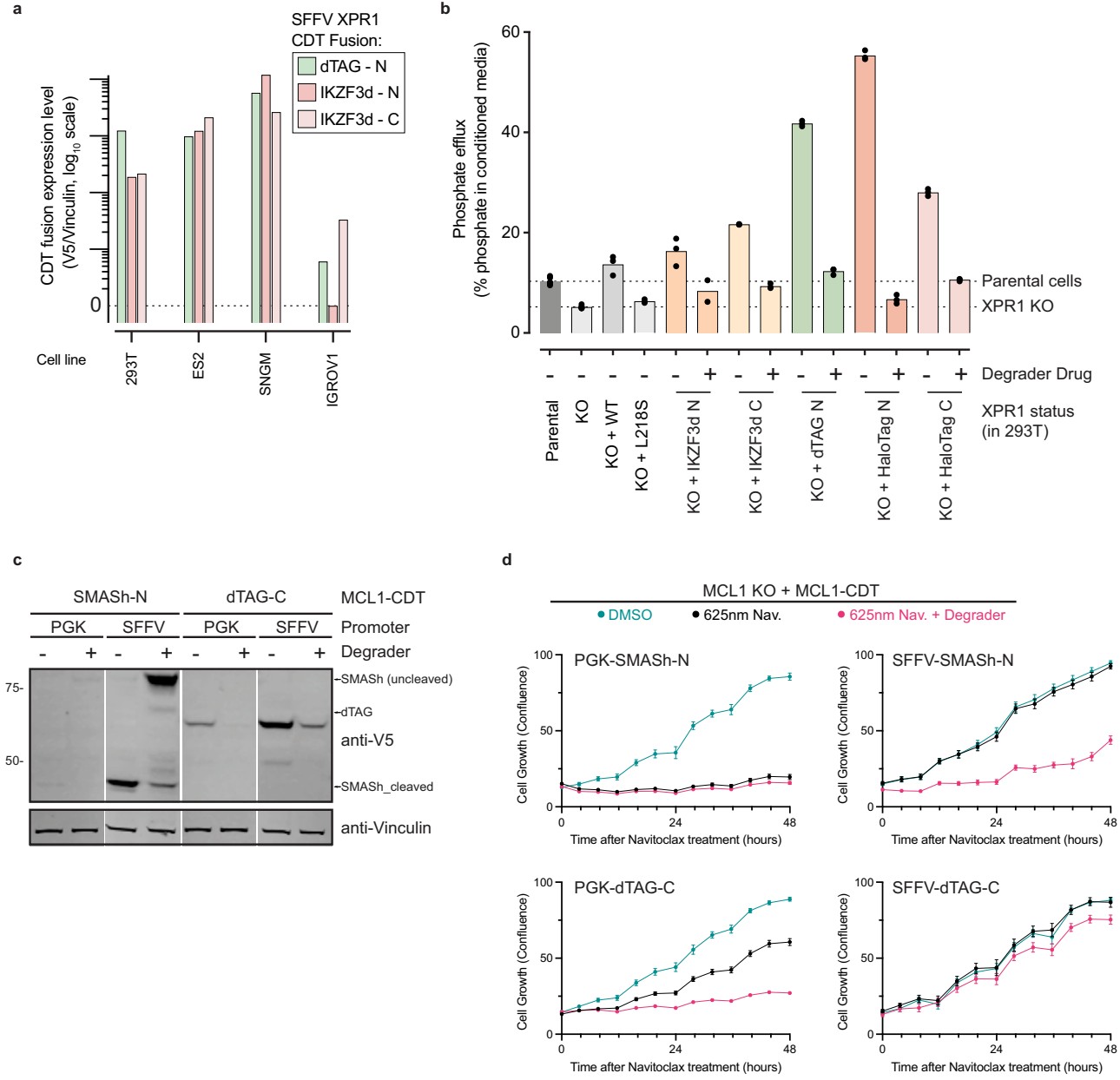

**Fig. 4 | Validation of functional activity for XPR1- and MCL1-CDT proteins. a** Comparison of expression levels for XPR1 degron fusion proteins in various cell lines. Expression was determined by quantifying the western blot V5 band intensity and normalizing it to a Vinculin loading control. **b** Phosphate efflux activity of XPR1 degron fusion proteins. 293T cells endogenously express XPR1, which was inactivated using CRISPR/Cas9 (XPR1-KO) followed by re-expression of wild-type XPR1 (WT), a hypomorphic allele (L218S), or the indicated SFFV-driven degron fusion proteins. Three days after the addition of degrader drug (1 µM Pomalidomide, 1 µM dTAG$^V$−1, or 1 µM HaloPROTAC3), phosphate efflux was measured by "loading cells" for 45 min with $^{32}PO_4{}^{3-}$, washing away any extracellular $^{32}P$, and then incubating the cells for 60 min and measuring the percentage of $^{32}P$ in the conditioned medium compared to cellular lysates. The bar height represents

the mean of technical triplicates (shown as points), and the results are representative of two independent experiments. **c** MCL1-CDT fusions are expressed and degraded in A375 cells. Cells expressing the indicated MCL1-CDT proteins were treated with 1 µM Asunaprevir or 1 µM dTAG$^V$−1 prior to evaluating protein levels by western blot. **d** Evaluation of MCL1 degron fusions to protect cells from Navitoclax-induced cell death. Endogenous *MCL1* was inactivated in A375 cells expressing the MCL1-CDT fusions shown in **c** and then pretreated with the indicated degrader drugs for 5 days (Asunaprevir) or 1 day (dTAG$^V$−1). At "time 0", the cells were treated with 625 nM Navitoclax and cell growth was evaluated with live-cell imaging, and confluency was evaluated through image analysis. Error bars represent the mean of $N = 3$ technical replicates and are representative of $N = 2$ independent experiments.

degradation is insufficient to phenocopy loss of the protein (Fig. 5a). While our studies were in the context of exogenous overexpression, we expect that insertion of a CDT at the endogenous locus for a given gene will likely have similar impacts on the expression of the gene product, although endogenous regulation may lessen the dramatic differences in expression that we observed. Indeed, here too, assessing multiple CDT fusions in parallel ensures higher chances of success.

Taken together, our data help to inform the development of a CDT strategy for target validation studies (Fig. 5b). We provide here a lentiviral vector system and a detailed Standard Operating Protocol (Supplementary Note 1) to quickly generate and test multiple CDTs in parallel to facilitate this process and highlight both successful and unsuccessful examples, the latter of which often go unpublished. Up-front testing of the five CDTs presented here enables the efficient identification of properly expressed and

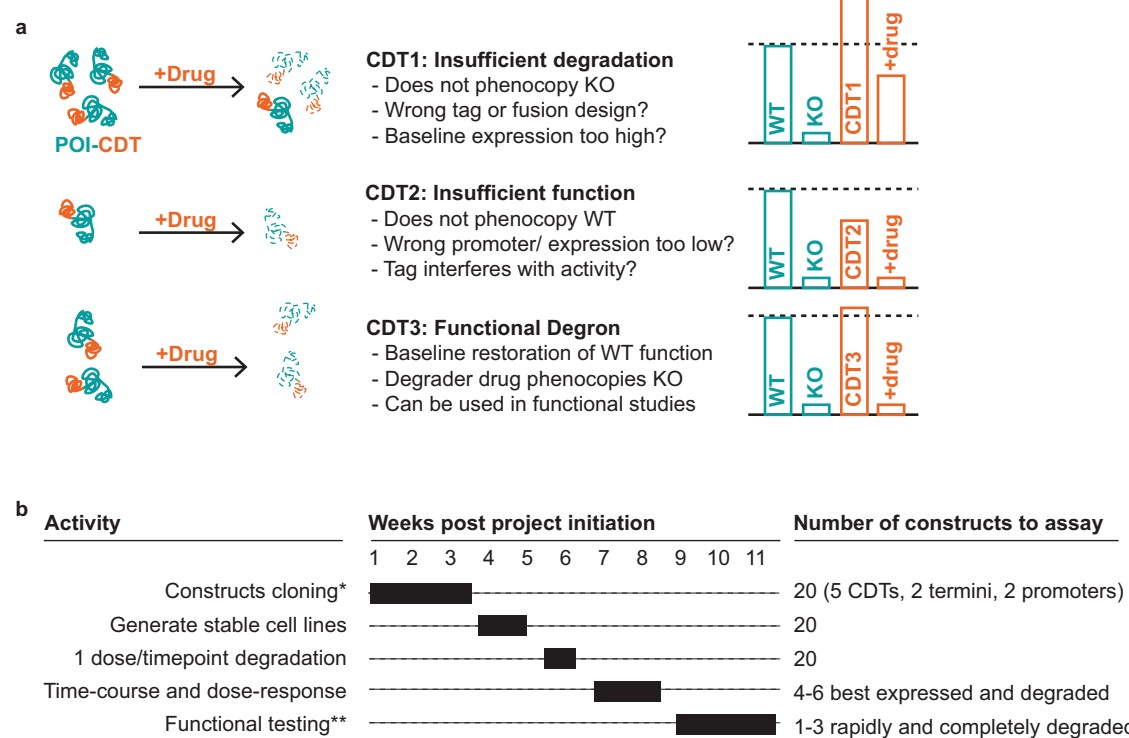

**Fig. 5 | A systematic approach to developing CDT fusions for target validation studies. a** Schematic summary of functional consequences of different expression levels and degradation potencies. In this example, CDT1 is a degron fusion protein in which degradation is not sufficiently potent to phenocopy inactivation of the target gene; CDT2 is either hypomorphic or not expressed at high enough levels to functionally replace the endogenous gene; CDT3 is the ideal situation where the degron fusion protein phenocopies both the expression of the endogenous gene and inactivation of the gene upon degrader drug addition. **b** A proposed timeline for evaluating the expression, degradation, and function of CDT fusions. Although 20 constructs can all be tested in parallel using the vectors and protocols presented here, priority can be given to particular CDTs if required (see Discussion). In most cases, a degraded and functionally relevant CDT fusion protein can be developed within 11 weeks. *Construct cloning via Contract Research Organization; **Functional testing is highly target-dependent and 3 weeks is an estimation.

degraded CDT fusions without burdensome and time-intensive iterations. If resources are limited, our work suggests the following ranking, from most to least robust: dTAG, SMASh, IKZF3d, HaloTag, and AID/AID2. This ranking takes into consideration degradation efficiency, baseline expression, absence of additional protein products (commonly seen with HaloTag-N), bio-orthogonality, and kinetics of degradation.

Future work will focus on developing CDT technologies for in vivo work. In Supplementary Table 1, we summarize prior pharmacokinetic (PK) studies for some of the small molecules analyzed here (dTAG-13[11], dTAG$^V$−1[11], Pomalidomide[41], and Asunaprevir[42]). We could not find similar data for the 5-Ph-IAA or HaloPROTAC3, although these compounds have been previously used in vivo[8,43]. Indeed, a systematic study of the PK and pharmacodynamic properties of each degrader drug would be beneficial to robustly assess in vivo performance across CDTs. We note that dTAG$^V$−1 has a reported half-life of 4.4 h when administered intraperitoneally and has been used in several in vivo studies. The development of better tool compounds for multiple CDTs with optimized PK properties would enable in vivo target validation studies where the dTAG system is not suitable for a particular POI (e.g., for XPR1, where the AID2 system performs better than dTAG). These studies would enable preclinical efficacy and toxicology studies following the degradation of a POI prior to the development of an optimized inhibitor, thus providing a robust assessment of their relevance as therapeutic targets.

## Methods
### Statistics and reproducibility
Details on statistical tests used in this study are contained in the methods for the relevant experiments. Statistical significance was calculated based on the distribution of biological replicates as opposed to technical or experimental replicates. All experiments were repeated at least three times to ensure reproducibility. No statistical method was used to predetermine the sample size. No data were excluded from the analyses. The experiments were not randomized. The Investigators were not blinded to allocation during experiments and outcome assessment.

### Construct design and cloning
All vectors included a pGK or an SFFV promoter driving the expression of a puromycin selection cassette, a p2a self-cleaving peptide, and the specific CDT (as shown in Fig. 1b). A rigid linker separated the CDT from the POI in the dTAG, the IKZF3d, and the HaloTag constructs. The HaloTag sequence was obtained from Craig Crews' lab. The SMAShTag ORF was obtained from Addgene. The dTAG ORF was obtained from Nathaniel Gray's lab. The IKZF3 minidegron sequence was obtained from Ben Ebert's lab. The AID and TIR1 ORF were obtained from Dr Johannes Zuber. A BamH1 unique cloning site was included at the N or C terminal of the V5 and sequences. All constructs were generated at Epoch Life Sciences and validated in-house by cloning the NanoLuc ORF. All constructs and vector maps are available on Addgene (#185760-185779). The deposited vectors express GFP-CDT fusion; the GFP insert can be excised via EcoR1/BamH1 digestion for directional cloning of different ORFs.

### Compounds
The compounds used in this study can be found in Table 1. dTAG$^V$−1 was either purchased commercially or synthesized in-house (see Supplementary Note 2). In a head-to-head comparison, there was no difference in the activity of dTAG$^V$−1 from these different sources.

## Table 1 | Compounds used in this study

| Compound | Supplier | Cat# |
|---|---|---|
| dTAG-13 | Tocris | 6605 |
| dTAG$^V$–1 | Tocris | 6914 |
| 3-Indoleacetic acid (IAA) | Sigma-Aldrich | 12886-5 G |
| 5-Phenyl IAA | MedChemExpress | HY-134653 |
| Asunaprevir | Ambeed Inc. | A542126 |
| Pomalidomide | Tocris | 6302 |
| HaloPROTAC3 | Promega | CS2072A01 |
| Navitoclax | Selleck Chemicals | S1001 |

## Table 2 | Antibodies used in this study

| Antigen | Supplier | Cat# | WB Dilution |
|---|---|---|---|
| V5 | Cell Signaling | D3H8Q | 1:200 (ICW) 1:2000 (WB) |
| CellTag 800 | LI-COR | #926-41090 | 1:500 (ICW) |
| Anti-Mouse 780 | LI-COR | #926-68070 | 1:1000 (ICW) 1:5000 (WB) |
| Anti-Rabbit 800 | LI-COR | #926-32211 | 1:1000 (ICW) 1:5000 (WB) |
| Vinculin | Sigma | V9131 | 1:5000 (WB) |
| KIDINS220 | Protein Tech | 21856-1-AP | 1:2000 (WB) |
| XPR1 | Sigma | HPA016557 | 1:2000 (WB) |

All compounds were dissolved in DMSO and were typically used at a final DMSO concentration of <0.1%.

### Antibodies
Table 2 contains the dilutions and catalog numbers of all antibodies used in this study.

### Cell culture
HEK-293T was procured from ATCC (CRL-3216). All other cell lines were procured by the Cancer Cell Line Encyclopedia and distributed for our use. The original sources of the cell lines were ATCC (MIA-PACA2, CRL-1420; A375, CRL-1619; HCT116, CCL-247; and ES2, CRL-1978), MilliPore Sigma (IGROV1, SCC203), or the Japanese Cell Resources Bank (SNGM, IFO50313). Cell lines were grown according to the manufacturer's instructions in Dulbecco's modified Eagle's medium (DMEM, Life Technologies) supplemented with 10% FBS (HEK-293T, MIA-PACA2, A375, and HCT116) or RPMI 1640 supplemented with 10% FBS (HEK-293T, SNGM, IGROV1, and ES2). HEK-293T expressing pLX-TRC313-TIR1$^{OS}$ were generated by the lentiviral introduction (see below) of the TIR1 open reading frame and selection with 300 μg/mL of hygromycin. AID2 fusions were assessed by introducing the F74G mutation into the TIR1 expression plasmid and generating an independent cell line expressing the mutant; expression levels of WT and F74G were confirmed to be similar by immunoblot. Expression of all CDT constructs was selected with 2 μg/mL puromycin.

### Lentivirus production
In a 24-multiwell plate, $1 \times 10^5$ HEK-293T cells per well were plated in 0.5 mL of media. Twenty-four hours after plating, transfection mixtures were made containing 50 μL Opti-MEM (Life Technologies), 250 ng packaging plasmid (psPAX2), 250 ng CDT plasmid, 25 ng envelope plasmid (pMD2.G), and 1.5 μL TransIT-LT1 (Mirus Bio). Transfection mixtures were incubated for 30 min at room temperature and added dropwise to cells. Twenty-four hours after transfection, the media was aspirated and replaced. Viral collections were then performed 24 and 48 h after media replacement.

### Stable cell line generation
In total, $1 \times 10^5$ cells were plated per well of a 24-well plate. About 200 μL degron virus was added to each well with polybrene (Santa Cruz Biotechnology) to a final concentration of 8 μg/mL and cells were returned to the incubator. 24 h after infection, the cells were selected with the relevant antibiotic.

### sgRNA
MCL1 – 5′  AGGCGCTGGAGACCTTACGA 3′
XPR1 – 5′  TCTGCAGCAGGATTAGACTG 3′

### Immunoblotting (IB)
Cells were collected and lysed in RIPA buffer supplemented with protease and phosphatase inhibitors (Roche). Protein concentrations were quantified, and equal amounts of protein were diluted with sample buffer, boiled, and loaded on Bis-Tris gels (NuPAGE). Gels were dry-transferred to nitrocellulose membrane (iBlot system, Life Technologies) and then probed with the indicated antibodies diluted in Intercept blocking buffer (Li-COR) overnight. Bands were detected using a Li-COR Odyssey CLX instrument, and bands were quantified using Image Studio.

### In-cell western (ICW)
About 20,000–40,000 cells were plated in 100 μL of growth media in black-walled 96-well plates with transparent bottom. Twenty-four hours later, a Tecan D300e Digital Dispenser was used to dispense drugs. To fix cells, 100 μL ice-cold methanol (Sigma-Aldrich) was slowly added to the side of the wells and the plates were incubated at room temperature for 10 min with gentle shaking. The methanol was removed, and the plates were washed five times with 200 μl of 0.1% NP40 buffer in PBS. About 50 μL of anti-V5 antibody was added, and the plates were then incubated overnight at 4 °C with gentle rocking. The next day, the plates were washed five times with 200 μl of 0.1% NP40 in PBS, then incubated for 1 h at room temperature in 50 μl of anti-Rabbit secondary and 50 μl of CellTag (Licor #926-41090). The plates were then washed three times with 200 μL 0.1% NP40 in PBS, the liquid was removed, and the plate was imaged on an LI-COR Odyssey imaging system. Quantification was performed using Image Studio.

### Washout experiments
Cells were plated in a 10 cm dish and treated with the indicated compound. Twenty-four hours after treatment, cells were washed with PBS, trypsinized, and replated in a drug-free medium for the indicated times in six-well plates. POI-CDT levels were determined by IB.

### In-cell luciferase assays
To determine luciferase activity of RFluc-CDT fusion proteins, cells growing in 96-well plates were treated with growth medium + 150 μg/mL D-Luciferin (Thermo Fisher) and luminescence was measured immediately (Envision Plate reader, Perkin Elmer). The culture medium was then replaced and cells were allowed to grow.

### Transient transfection and degradation
Twenty-four hours after plating, HEK-293T cells were transiently transfected with differing amounts of the indicated vectors. Twenty-four hours later, the cells were replated, and 24 h after that, the cells were treated with DMSO or 1 μM dTAG$^V$–1. Twenty-four hours after treatment, PRMT5-CDT levels were assessed by WB.

### XRBD protein purification
We kindly thank Jean Luc Battini for providing the sequence for the XRBD-mFc construct published previously[29,30]. The plasmid encoding XRBD (strain NZB) was synthesized and cloned into pcDNA3.4 by Thermo Fisher Scientific GENEART GmbH. XRBD protein was

expressed in CHO cells and purified using Protein A affinity chromatography by Thermo Fisher Scientific GENEART GmbH.

## XRBD flow cytometry

Cells were lifted from culture vessels using TrypLE Express (Thermo Fisher cat#12604013) and then diluted in PBS + 2% FBS. About 300,000 cells were transferred to a U-bottom 96-well plate in 50 uL, followed by the addition of 50 uL of a 100 nM XRBD staining solution. Cells were incubated at 37 °C for 40 min, washed once, and then incubated with an anti-mouse secondary antibody conjugated to AlexaFluor488 (ThermoFisher cat# A-11004). Cells were then incubated on ice for 40 min, washed four times in PBS + 2% FBS, and then analyzed on a CytoFlex LX instrument. At least 10,000 single cell events were recorded for each condition (see Supplementary Fig. 12b for representative gating strategy).

## Phosphate uptake and efflux assays

To determine phosphate efflux, HEK-293T cells stably expressing the indicated CDT fusions were plated in poly-L-Lysine coated 24-well plates and treated with the indicated compound for 72 h. Cells were first "pulsed" using "no phosphate" RPMI 1640 supplemented with 10 µci/mL $^{32}PO_4$ (Perkin Elmer NEX053001MC) and incubated at room temperature for 60 min. The cells were then washed with "no phosphate" RPMI 1640. The "chase" of phosphate efflux was then measured by incubating cells in high-phosphate RPMI (i.e., standard RPMI 1640) for another 45 min. The conditioned medium was collected, and the cells were washed three times in "no phosphate" RPMI 1640. Cells were then lysed in 1% Triton X-100, and the amount of $^{32}P$ in the lysate and conditioned medium were measured via a scintillation counter. The extent of phosphate efflux was determined by dividing the $^{32}P$ measured in the conditioned medium by the total $^{32}P$ measured for that sample (in cell lysates and in the conditioned medium).

## Viability assays to assess Navitoclax sensitivity with MCL1-CDTs

A375 cells were pretreated with 1 µM Asunaprevir for 4 days prior to re-plating the cells in a 96-well plate (with continued exposure to Asunaprevir and co-treatment with 1 µM dTAG$^V$−1). Twenty-four hours later, the cells were treated with either DMSO or 625 nM Navitoclax, as indicated. Cell confluency was then assessed using live-cell imaging every 4 h using an Essen Incucyte S3 Live-Cell Analysis Instrument.

## Reporting summary

Further information on research design is available in the Nature Research Reporting Summary linked to this article.

## Data availability

All data associated with this manuscript are available in the Supplementary Information file or as Source Data, which is provided with this manuscript. All constructs and vector maps are available on Addgene (#185760-185779). Source data are provided with this paper.

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

## Acknowledgements
This work was funded in part by Novo Ventures and grants from the National Cancer Institute CA242457 (T.R.G.), CA212229 (D.P.B.), and CA233626 (W.R.S.). D.M. performed ICW and WB of the STAG1 and STAG2 CDT fusions and was supported by a Novartis DDTRP grant to Z.T. We thank D Tennen, B Buckley, M Veneskey, B Hart, and M Cecilia Saberi for technical support.

## Author contributions
D.P.B. and A.I. conceived of the "degron panel" approach and designed the lentiviral vectors. Z.M.-B. cloned the initial NanoLuc constructs and validated the approach. D.P.B., Z.M.-B., S.O., T.A.S., N.B., M.C., A.A.G., J.K., C.L., D.M., K.T., S.J.W., and A.I. evaluated CDT efficiency across targets and conducted additional experiments. T.C.A. synthesized dTAG$^V$–1. D.P.B., B.R.P., F.V., Z.T., T.R.G., W.R.S., and A.I. supervised the work. D.P.B. and A.I. wrote the manuscript. D.P.B., T.R.G., W.R.S., and A.I. edited the manuscript.

## Competing interests
F.V. receives research funding from Novo Ventures. T.R.G. has an equity interest in or receives consulting income from Sherlock Biosciences and Anji Pharmaceuticals and receives research funding from Calico Life Sciences, Bayer HealthCare, and Novo Holdings. A.I. receives cash compensation for consulting with Ridgeline Discovery. W.R.S. is a Board or SAB member and holds equity in Ideaya Biosciences, Civetta Therapeutics, Red Ridge Bio, and 2Seventy Bio and has consulted for Array, Astex, Epidarex Capital, Ipsen, PearlRiver Therapeutics, Merck Pharmaceuticals, Sanofi, Servier, and Syndax Pharmaceuticals and receives research funding from Pfizer Pharmaceuticals, Merck Pharmaceuticals, Ideaya Biosciences, Calico, Boehringer-Ingelheim, Bristol Myers Squibb, and Ridgeline Discovery. W.R.S. is a co-patent holder on EGFR mutation diagnostic patents. The remaining authors declare no competing interests.
