## [Peer Review File · Nature Communications]

REVIEWER COMMENTS

Reviewer #1 (Remarks to the Author):

In this technical tour de force, Bondeson et al systematically study the efficacy of leading chemically inducible degradation (CID) domains across a variety of protein targets. The comparisons are technically well done and the results informative. I only have minor concerns about how certain results are presented or communicated.

1. The abstract states dTAG, IKZF3d, and HaloTag are the most robust CIDs. However Fig 2b shows HaloTag working with >95% efficacy only for 5 proteins, all when fused at the CT only. This is worse than SMASH. The discussion also states the most consistent CIDs were dTAG and IKZF3d with no mention of HaloTag. Thus the abstract should probably instead state dTAG and IKZF3d are the most robust CIDs, and SMASH and HaloTag work well for a smaller fraction of the proteins tested, and AID generally works poorly. If that is too long, the abstract can just state dTAG and IKZF3d are the most robust CIDs.

2. The immunofluorescence quantitation used in Fig 1C is called an "in-cell western (ICW)" (page 2, last paragraph). The defining feature of a western is size assessment by migration, not immunodetection. That is not present in this immunofluorescence experiment, so to call it a western is misleading. There is no need to rename immunofluorescence as western anyway. It already has a name: immunofluorescence.

3. On page 3, paragraph 2, in "We generated HEK-293T cells stably expressing each POI-CID fusion..." it would be useful to write instead "We generated polyclonal HEK293T cells stably expressing each POI-CID fusion from the X promoter" (X should be replaced with the actual promoter)

4. On page 3, paragraph 4, the reference to Figure 1c appears to be incorrect.

5. Figure 1c may be easier to understand if, in the lower chart, percent remaining is plotted instead of percent degraded. It will be easier to understand when relating to expression levels.

6. Figure 2a may be easier to understand if the LUT is rainbow showing a spectrum of colors rather than just two colors. It is especially difficult to discern any differences between 75% and 100%, or between 25% and 0%, with the current LUT.

7. Figure 2b lower table really should list N fusions and C fusions separately with all N fusions followed by all C fusions. This is because it's clear from Figure 2a that for each CID, NT fusions tend to behave similarly across proteins, and CT fusions tend to behave similarly across proteins, while NT and CT fusions for one protein can be very different. That is, CID function is more usefully grouped by position than by target. In addition, the current listing does not make clear the total generalizability of each CID.

Reviewer #2 (Remarks to the Author):

Bondeson et al describe a systematized approach to evaluate several degradation systems that have emerged over the last few years in a head-to-head approach. Overall, this is a useful report and making the materials publicly available will certainly benefit the community. I would support publication eventually but have a few points that I think the authors should consider. I apologize to the authors if my feedback delayed their review process.... COVID.

1. For nearly 3 decades, CID has been used to stand for chemically induced dimerization. It is unnecessary to confuse this by now coining chemically inducible degrons as CID. This field has enough trouble defining and categorizing things like molecular glues and PROTACs.

2. Sometimes the degrader dose needs to be dialed in for difficult target-compound pairs. I think a full dose responses for unsuccessful examples would be worthwhile to rule out this a source of variability.

3. I would caution against the inference that high expression overwhelms the degradation machinery. For example, an alternative explanation would be that high expression from a transfected construct within a compressed period of time may overwhelm the protein folding machinery and result in an improperly folded protein that does not bind the ligand.

4. There is acknowledgement of the several studies that have shown degron systems don't impair Luc activity, but nothing acknowledging the many examples when these systems don't impact the function of cancer-relevant POIs. This dates back to the earliest report of the dTAG system from the Bradner group studying ENL (Nature 2017) and leads up to the Stegmaier group's recent work with NXT1 (cancer discovery 2021) and TRIM8 (cancer cell 2021). This is important context for future readers that will approach your paper as a resource for their own experiments.

5. I have seen dTAG spelled as dTag in the manuscript. For example, “dTag13” is used in Ext Fig 2. I believe the Bradner group intended for this to be “dTAG” based on their first description of the “dTAG-13” molecule (Erb Nature 2017). This is important for consistency and to not confuse newcomers to the field.

6. It is interesting that XPR1 only failed to express in the cell line that requires it for survival. This looks to me like it could be acting as a dominant negative and selection for transduced cells selects for low XPR1 expression. This may indicate that the IKZF and FKBP tags disrupt function (potentially distinct from the phosphate efflux that is studied later). I would recommend testing with the AID and AID2 systems to see if that tag has better performance in IGROV1.

7. I would prefer a stronger statement highlighting that this study uses an older version of AID, since the very challenges your study encountered have been previously encountered and acknowledged and significant work has gone into addressing them (Li Nat Methods 2019; Sathyan Gene Dev 2019; Yesbolatova Nat Comm 2020).

8. Finally, I wish knock-ins could be included for head-to-head comparisons because I think there will be major differences in expression among protein tags when constrained by the endogenous promoter. However, I can appreciate if the authors feel that this is beyond the scope of the current study.

Reviewer #3 (Remarks to the Author):

In this manuscript submitted to Nature Communications, Bondeson and colleagues examine certain performance characteristics of experimental methods to temporally control protein stability. This is a timely article, as the number of methods utilized has recently expanded. The study of five degradation systems for 16 proteins is a nice, and potentially instructive, survey. While I would hesitate to draw conclusions on 16 protein targets, for the reasons listed below notably including thin coverage of membrane proteins, the study is of interest and could be instructive as to what *not* to reach for first in a degradation system. This is a well controlled study that is appropriate for publication in Nature Communications. Many will be as interested to read the work, as I was today.

Comments are listed below, but I invite the authors to include design limitations when selecting a "first" degradation system to test, as I curiously arrive at a different conclusion than the authors awash in all these data. The biological performance data support that "the most consistently degraded CIDs were the IKZF3d and dTAG systems", as mentioned in the discussion. But I do not see expression level as a challenge to the experimental biologist, who will surely titrate expression vectors for optimal expression (especially with two promoters here to choose from). Rather, the restricted utility of IKZF3d in only human (and cyno) model systems, combined with the complicating degradation of a large number of other proteins (IKZF1, IKZF3; situationally also CK1a, SALL4, etc.), renders this degron strongly disfavored for mechanistic biology. Having read this study, I would sooner try dTAG, and if this did not work reach for the HaloTag system next. Both are fully orthogonal to human biology, key for mechanistic research. Both also can adapt well to the murine model system, where so much important biology is studied. For this reason, I would recommend a comparison of the three "finalists" in a suitably "fair" in vivo setting. This experiment could be with luciferase in a human xenograft, for example. Overall, this is an excellent paper worthy of publication. The conclusion, however, that dTAG, IKZF3d and HaloTag are "the most robust" as in the abstract strikes me as imprecise guidance to the newcomer. My sense is that dTAG has only the challenge of not degrading XPR1, and would out-perform the others in vivo. Amazing work!

[Full disclosure, my prior academic laboratory contributed the dTAG system but did not remotely validate the system to this extent. But we also innovated an IKZF tag system here at NIBR, which we do not recommend nearly as much as dTAG for all the above reasons.]

Comments:

1. "CID" is not ideal as an acronym, especially here, where in the field of bifunctionality the term originated and is still used to denote "Chemical Inducers of Dimerization". Also, the degron is not "induced" by the chemical, rather degradation is, so the term strikes me as also imprecise. I am not an authority on naming conventions, but would think that "Chemically-Responsive Degradation" or something similar would be more accurate.

2. The lentiviral vectors will be a valuable resource to the community.

3. The requirement of co-expression of the Tir1 E3 ligase is an additional unnecessary complexity, as is the low potency of the chemical required. This renders use in vivo particularly disadvantageous. Clarity on this for the uninitiated experimentalist is welcome.

4. Two challenges with the IKZF3d tag are not discussed. First, is the degradation of Ikaros-1 and Ikaros-3 transcription factors by the system in human cells. Compared to HaloTag and dTAG, which are bio-orthogonal solutions, this is a particular disadvantage in mechanistic cell biology. Also, the IKZF3d system

will not work in murine models, owing to the lack of murine Cereblon to degrade substrate proteins. Clarity on these negative differentiators for the uninitiated experimentalist is again welcome.

5. The interpretation of the kinetic data in Figure 3a, when comparing proteins, should include a consideration of protein resynthesis rate, not only the "interaction with the degradation machinery".

6. I am pleasantly surprised by the rapid degradation of the Halotag degron, and admittedly unpleasantly surprised by the rapid recovery with wash-out. Why would a covalent strategy have rapid loss of signal?

7. Sustained, suppressed degradation in vitro can be a contributor to sustained effect in vivo. The only missing experiment in this study is a comparison of in vivo performance with the ligands. There is surely a chance that the ligands will not work in murine models, as for IKZF3d above. So fair comparison might require pharmacodynamic loss of luciferase in a human xenograft, optimally expressed as a fusion directed by the data in Figure 1, and Supplemental Figure 1.

Congratulations, all. Best wishes, Jay Bradner

Response to reviewer comments

Reviewer #1 (Remarks to the Author):

In this technical tour de force, Bondeson et al systematically study the efficacy of leading chemically inducible degradation (CID) domains across a variety of protein targets. The comparisons are technically well done and the results informative. I only have minor concerns about how certain results are presented or communicated.

1. The abstract states dTAG, IKZF3d, and HaloTag are the most robust CIDs. However Fig 2b shows HaloTag working with >95% efficacy only for 5 proteins, all when fused at the CT only. This is worse than SMASh. The discussion also states the most consistent CIDs were dTAG and IKZF3d with no mention of HaloTag. Thus the abstract should probably instead state dTAG and IKZF3d are the most robust CIDs, and SMASh and HaloTag work well for a smaller fraction of the proteins tested, and AID generally works poorly. If that is too long, the abstract can just state dTAG and IKZF3d are the most robust CIDs.

At the suggestions of reviewers 1 and 3, we have modified the overall interpretation of which Conditional Degron Tag is the most robust across targets. We have highlighted that dTAG, SMASh, and IKZF3d achieve the most robust degradation across targets and have included important caveats about each technology. The data continue to support the importance of evaluating multiple systems in parallel, and if possible, that investigators test multiple strategies in parallel as the most efficient mechanism by which to identify an optimal strategy.

2. The immunofluorescence quantitation used in Fig 1C is called an "in-cell western (ICW)" (page 2, last paragraph). The defining feature of a western is size assessment by migration, not immunodetection. That is not present in this immunofluorescence experiment, so to call it a western is misleading. There is no need to rename immunofluorescence as western anyway. It already has a name: immunofluorescence.

The term "In-cell Western" has been used commercially (e.g., Li-Cor) and in the literature for some time. Notably, a Pubmed search for the quoted term "In-cell western" returned 202 citations where it is primarily used to indicate an antibody-based, quantitative measurement of a bulk population of cells. Given the prevalence of this term, particularly in the targeted protein degradation field, we prefer to maintain the use of ICW.

3. On page 3, paragraph 2, in "We generated HEK-293T cells stably expressing each POI-CID fusion..." it would be useful to write instead "We generated polyclonal HEK293T cells stably

expressing each POI-CID fusion from the X promoter" (X should be replaced with the actual promoter)

We have made this change.

4. On page 3, paragraph 4, the reference to Figure 1c appears to be incorrect.

We had intended to point the reader to the differences in degradation for NanoLuc CDT-fusions when expressed from PGK or SFFV promoters, and have now further clarified this point in the text.

5. Figure 1c may be easier to understand if, in the lower chart, percent remaining is plotted instead of percent degraded. It will be easier to understand when relating to expression levels.

At the Reviewer's suggestion, we have changed Figure 1c to the percent of protein remaining.

6. Figure 2a may be easier to understand if the LUT is rainbow showing a spectrum of colors rather than just two colors. It is especially difficult to discern any differences between 75% and 100%, or between 25% and 0%, with the current LUT.

We evaluated a spectrum of different colors for Figure 2a, which we found to be confusing and difficult to interpret. To improve the readability of the graph, we now present the "Percent Degradation" value encoded by a two-color, discrete (0-10%, 11-20%, etc.) look-up table. The LUT now enables a reader to easily evaluate differences in the 0-25% as well as the 75-100% ranges.

7. Figure 2b lower table really should list N fusions and C fusions separately with all N fusions followed by all C fusions. This is because it's clear from Figure 2a that for each CID, NT fusions tend to behave similarly across proteins, and CT fusions tend to behave similarly across proteins, while NT and CT fusions for one protein can be very different. That is, CID function is more usefully grouped by position than by target. In addition, the current listing does not make clear the total generalizability of each CID.

In response to the reviewer's request, we have now grouped CDT fusions by N- and C-terminus. "Figure 2b lower table" is now Figure 2d.

Reviewer #2 (Remarks to the Author):

Bondeson et al describe a systematized approach to evaluate several degradation systems that have emerged over the last few years in a head-to-head approach. Overall, this is a useful report and making the materials publicly available will certainly benefit the community. I would

support publication eventually but have a few points that I think the authors should consider. I apologize to the authors if my feedback delayed their review process.... COVID.

1. For nearly 3 decades, CID has been used to stand for chemically induced dimerization. It is unnecessary to confuse this by now coining chemically inducible degrons as CID. This field has enough trouble defining and categorizing things like molecular glues and PROTACs.

We thank reviewers #2 and #3 for their comments on how to properly situate this work within the broader scientific community. In response, we have chosen to use the term “Conditional Degron Tags”, or CDT to avoid the confusion with the use of the CID term in dimerization.

2. Sometimes the degrader dose needs to be dialed in for difficult target-compound pairs. I think a full dose responses for unsuccessful examples would be worthwhile to rule out this a source of variability.

In response, we have examined a full dose response for five targets: NanoLuc, PRMT5, RFLuc, WSB2, and XPR1. These data indicate fairly consistent behavior across doses. Thus, it is unlikely inadequate drug exposure accounts for lack of degradation in the unsuccessful examples.

3. I would caution against the inference that high expression overwhelms the degradation machinery. For example, an alternative explanation would be that high expression from a transfected construct within a compressed period of time may overwhelm the protein folding machinery and result in an improperly folded protein that does not bind the ligand.

We have edited the text to acknowledge the possibility of additional biological mechanisms that might prohibit degradation of highly overexpressed proteins, including misfolding.

4. There is acknowledgement of the several studies that have shown degron systems don't impair Luc activity, but nothing acknowledging the many examples when these systems don't impact the function of cancer-relevant POIs. This dates back to the earliest report of the dTAG system from the Bradner group studying ENL (Nature 2017) and leads up to the Stegmaier group's recent work with NXT1 (cancer discovery 2021) and TRIM8 (cancer cell 2021). This is important context for future readers that will approach your paper as a resource for their own experiments.

We thank the Reviewer for highlighting this important context, and have added references to the indicated papers.

5. I have seen dTAG spelled as dTag in the manuscript. For example, “dTag13” is used in Ext Fig 2. I believe the Bradner group intended for this to be “dTAG” based on their first description of the “dTAG-13” molecule (Erb Nature 2017). This is important for consistency and to not confuse newcomers to the field.

We thank the Reviewer for pointing this discrepancy out, and have corrected all instances to ‘dTAG.’

6. It is interesting that XPR1 only failed to express in the cell line that requires it for survival. This looks to me like it could be acting as a dominant negative and selection for transduced cells selects for low XPR1 expression. This may indicate that the IKZF and FKBP tags disrupt function (potentially distinct from the phosphate efflux that is studied later). I would recommend testing with the AID and AID2 systems to see if that tag has better performance in IGROV1.

In response, we attempted to overexpress XPR1-AID-C fusion proteins in IGROV1, but found very poor expression of these constructs in IGROV1 but not in 293T. These data highlight that overexpression of a given POI may not be possible in particular cell line contexts, an important note of caution to prospective users of the technologies.

7. I would prefer a stronger statement highlighting that this study uses an older version of AID, since the very challenges your study encountered have been previously encountered and acknowledged and significant work has gone into addressing them (Li Nat Methods 2019; Sathyan Gene Dev 2019; Yesbolatova Nat Comm 2020).

To address this issue we have evaluated the AID2 system across 9 different POI, and have included this new data in Figure 2. These data indicate that AID2 does provide some improvement in basal POI-AID2 fusion proteins and more potent degradation with the improved compound consistent with the noted references. In most cases however, this improvement was marginal and did not reach the >90% degradation efficiency. In addition to the inclusion of this data in Figure 2, we have changed the references to AID1 and AID2 and added the recommended citations.

8. Finally, I wish knock-ins could be included for head-to-head comparisons because I think there will be major differences in expression among protein tags when constrained by the endogenous promoter. However, I can appreciate if the authors feel that this is beyond the scope of the current study.

To systematically study CDTs across multiple POI, we chose to remove the endogenous promoter as a source of variability, but agree with the Reviewer that the observed differences in expression might be resolved when expression of the

POI-CDT fusions is constrained by endogenous transcriptional regulation and have modified the discussion to reflect this.

Reviewer #3 (Remarks to the Author):

In this manuscript submitted to Nature Communications, Bondeson and colleagues examine certain performance characteristics of experimental methods to temporally control protein stability. This is a timely article, as the number of methods utilized has recently expanded. The study of five degradation systems for 16 proteins is a nice, and potentially instructive, survey. While I would hesitate to draw conclusions on 16 protein targets, for the reasons listed below notably including thin coverage of membrane proteins, the study is of interest and could be instructive as to what *not* to reach for first in a degradation system. This is a well controlled study that is appropriate for publication in Nature Communications. Many will be as interested to read the work, as I was today.

Comments are listed below, but I invite the authors to include design limitations when selecting a "first" degradation system to test, as I curiously arrive at a different conclusion than the authors awash in all these data. The biological performance data support that "the most consistently degraded CIDs were the IKZF3d and dTAG systems", as mentioned in the discussion. But I do not see expression level as a challenge to the experimental biologist, who will surely titrate expression vectors for optimal expression (especially with two promoters here to choose from). Rather, the restricted utility of IKZF3d in only human (and cyno) model systems, combined with the complicating degradation of a large number of other proteins (IKZF1, IKZF3; situationally also CK1a, SALL4, etc.), renders this degron strongly disfavored for mechanistic biology. Having read this study, I would sooner try dTAG, and if this did not work reach for the HaloTag system next. Both are fully orthogonal to human biology, key for mechanistic research. Both also can adapt well to the murine model system, where so much important biology is studied. For this reason, I would recommend a comparison of the three "finalists" in a suitably "fair" in vivo setting. This experiment could be with luciferase in a human xenograft, for example. Overall, this is an excellent paper worthy of publication. The conclusion, however, that dTAG, IKZF3d and HaloTag are "the most robust" as in the abstract strikes me as imprecise guidance to the newcomer. My sense is that dTAG has only the challenge of not degrading XPR1, and would out-perform the others in vivo. Amazing work!

[Full disclosure, my prior academic laboratory contributed the dTAG system but did not remotely validate the system to this extent. But we also innovated an IKZF tag system here at NIBR, which we do not recommend nearly as much as dTAG for all the above reasons.]

Comments:

1. "CID" is not ideal as an acronym, especially here, where in the field of bifunctionality the term originated and is still used to denote "Chemical Inducers of Dimerization". Also, the degron is not "induced" by the chemical, rather degradation is, so the term strikes me as also imprecise. I

am not an authority on naming conventions, but would think that "Chemically-Responsive Degradation" or something similar would be more accurate.

We have changed the text to use the term “Conditional Degron Tags”(CDT), to captures the primary goal of these technologies.

2. The lentiviral vectors will be a valuable resource to the community.

The vectors have been deposited at Addgene and we will make the vectors and a detailed SOP available upon publication.

3. The requirement of co-expression of the Tir1 E3 ligase is an additional unnecessary complexity, as is the low potency of the chemical required. This renders use in vivo particularly disadvantageous. Clarity on this for the uninitiated experimentalist is welcome.

We have added this important caveat on the use of AID systems in genetically engineered models.

4. Two challenges with the IKZF3d tag are not discussed. First, is the degradation of Ikaros-1 and Ikaros-3 transcription factors by the system in human cells. Compared to HaloTag and dTAG, which are bio-orthogonal solutions, this is a particular disadvantage in mechanistic cell biology. Also, the IKZF3d system will not work in murine models, owing to the lack of murine Cereblon to degrade substrate proteins. Clarity on these negative differentiators for the uninitiated experimentalist is again welcome.

To address the first issue, we have highlighted in the discussion that IMiD compounds are capable of degrading many additional substrates which can complicate the interpretation of phenotypic outcomes. To address the second caveat, we have expanded our prior analysis of CDT activity to include NIH-3T3 fibroblast cells. Here we show that all the tags work equivalently well in human versus murine contexts, except for the IKZF3d tag.

5. The interpretation of the kinetic data in Figure 3a, when comparing proteins, should include a consideration of protein resynthesis rate, not only the "interaction with the degradation machinery".

We have modified the discussion to include the potential for differences in the rate of resynthesis to be an important factor.

6. I am pleasantly surprised by the rapid degradation of the Halotag degron, and admittedly unpleasantly surprised by the rapid recovery with wash-out. Why would a covalent strategy have rapid loss of signal?

We hypothesize that the ligand covalently bound to HaloTag is not able to dissociate from degraded peptides and re-engage new substrate molecules (i.e. it is not catalytic). Thus, once excess HaloPROTAC3 is removed via washout, the protein is able to recover as quickly as its re-synthesis rate allows

7. Sustained, suppressed degradation *in vitro* can be a contributor to sustained effect *in vivo*. The only missing experiment in this study is a comparison of *in vivo* performance with the ligands. There is surely a chance that the ligands will not work in murine models, as for IKZF3d above. So fair comparison might require pharmacodynamic loss of luciferase in a human xenograft, optimally expressed as a fusion directed by the data in Figure 1, and Supplemental Figure 1.

The Reviewer requests an *in vivo* comparison of the different CDT technologies, which we suggest would require extensive compound formulation, pharmacokinetic evaluation, and pharmacodynamic analyses across five different compounds. In addition, as this manuscript seeks to evaluate CDT performance across multiple POI, assessing the degradation of only luciferase fusions will provide a limited view of the robustness of each CDT for *in vivo* use, as individual targets will require different schedule of administration. These studies are well beyond our *in vivo* capabilities/budgets. To aid readers interested in using CDT technologies *in vivo*, we have included an additional table that summarizes prior PK characterization of individual CDTs. Based on the reported PK properties of individual compounds and their activities in cellular systems that we present in this work, we recommend the dTAG system for researchers requiring *in vivo* studies. We have added a comment in the discussion to this effect.

Nevertheless, a full *in vivo* characterization across multiple CDTs and/or targets, while surely be of benefit to the community, is currently outside the scope of the current study.

REVIEWERS' COMMENTS

Reviewer #1 (Remarks to the Author):

The authors have satisfactorily addressed my concerns. This work will be a valuable resource to the community.

Reviewer #2 (Remarks to the Author):

In the revised manuscript, the authors have sufficiently addressed the points of my review from the initial submission.

Reviewer #3 (Remarks to the Author):

The authors have thoughtfully addressed all critical comments from my review, as well as the other two Reviewers (IMHO). I have no further comments relevant to this index comparative publication, and suggest proceeding with publication. Congratulations.